# Segment-Level Attribution for Selective Learning of Long Reasoning Traces

**Siyuan Wang, Yanchen Liu, Xiang Ren**
University of Southern California
{sw_641,liuyanch,xiangren}@usc.edu

## Abstract

Large Reasoning Models (LRMs) achieve strong reasoning performance by generating long chains of thought (CoTs), yet only a small fraction of these traces meaningfully contributes to answer prediction, while the majority contains repetitive or truncated content. Such output redundancy is further propagated after supervised finetuning (SFT), as models learn to imitate verbose but uninformative patterns, which can degrade performance. To this end, we incorporate integrated gradient attribution to quantify each token's influence on final answers and aggregate them into two segment-level metrics: (1) *attribution strength* measures the overall attribution magnitude; and (2) *direction consistency* captures whether tokens' attributions within a segment are uniformly positive or negative (high consistency), or a mixture of both (moderate consistency). Based on these two metrics, we propose a segment-level selective learning framework to identify important segments with high attribution strength but moderate consistency that indicate reflective rather than shallow reasoning. The framework then applies selective SFT on these important segments while masking loss for unimportant ones. Experiments across multiple models and datasets show that our approach improves accuracy and output efficiency, enabling more effective learning from long reasoning traces [1].

## 1 Introduction

Recent Large Reasoning Models (LRMs) (Jaech et al., 2024; Guo et al., 2025; Yang et al., 2025) have demonstrated strong capabilities in solving complex problems. Their effectiveness is largely attributed to test-time scaling by increasing computation during inference to produce extended chains of thought (CoT) (Wei et al., 2022) that include detailed problem understanding, step-by-step solution processes, and comprehensive verification. These long-CoT trajectories have also become valuable supervisory resources for cold-start supervised finetuning (SFT) (Muennighoff et al., 2025).

However, current reasoning CoTs often span thousands of tokens, of which only a small fraction meaningfully contributes to reaching the correct answer or improving confidence (Sui et al., 2025). A substantial portion consists of redundant repetition or incomplete truncated thoughts (Wang et al., 2025d), as illustrated in Fig. 1 (left). More critically, verbosity without substance actively degrade reasoning performance as CoT length increases (Wu et al., 2025; Huang et al., 2025). This phenomenon is also evident by the right-top panel of Fig. 1 where incorrect CoTs are typically correlated with more segments and tokens than correct CoTs for the same queries. Training LLMs on verbose CoT supervision with sparse positive contribution further exacerbates these issues. Models learn to imitate redundant behaviors, waste learning capacity on trivial continuations, and fail to prioritize the crucial high-impact parts of reasoning sequences Lin et al. (2024). As a result, finetuned models tend to achieve limited accuracy gains and generate inefficient outputs.

Prior studies have explored various strategies to identify important parts in long reasoning chains to construct compressed CoT supervision for efficiency purposes. However, they either focus on fine-grained token-level analysis (Xia et al., 2025b) that neglects semantic integrity and fails to interpret redundancy in terms of meaningful reasoning units, or rely on segment-level perplexity (Cui et al., 2025a) or entropy (Li et al., 2025b) calculations. These indirect metrics provide not entirely consistent

---

[1]Code and data are available at https://github.com/SiyuanWangw/SegmentSelectiveSFT.

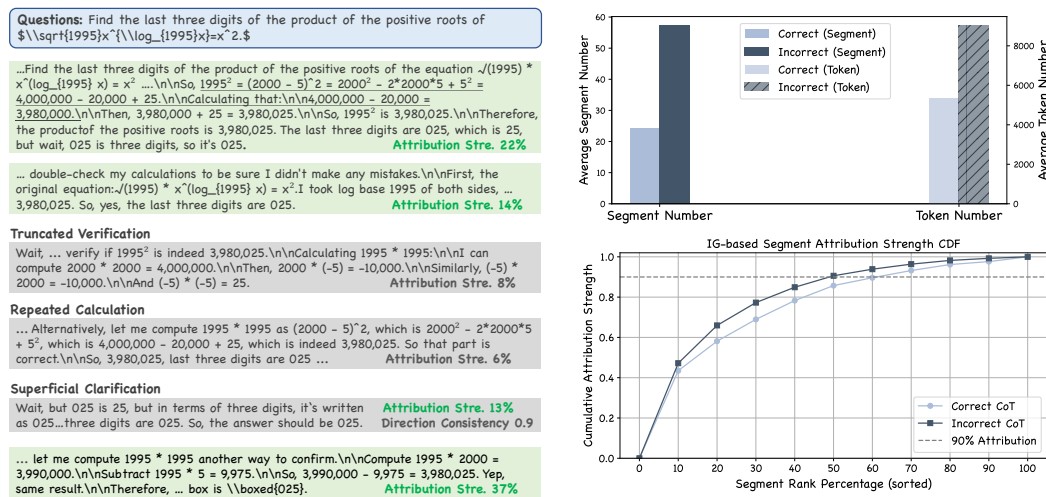

Figure 1: **Left**: An illustrative CoT with important (green blocks) and redundant segments (gray blocks). Our metrics distinguish important from redundant segments (repetitions, truncations, superficial clarifications) with low strength and extremely high consistency. "Attribution Stre." denotes normalized strength across all segments. **Right-top**: Segment and token counts in correct vs. incorrect CoTs for the same queries. **Right-bottom**: Cumulative distribution function (CDF) of normalized segment strength in correct and incorrect CoTs, with segment ordered in descending strength.

signals of importance and are prone to both false positives and false negatives. False positives occur when methods overemphasize superficial scaffolding text (e.g., "So, let's calculate step by step") that contributes little to actual reasoning yet serves as linguistic bridges whose removal disrupts subsequent textual coherence. False negatives arise when methods filter out independent verification or intermediate conclusions that, while exhibiting low-entropy and their removal not affecting linguistic fluency, significantly enhance the probability of reaching correct final answers. Consequently, existing methods cannot accurately and comprehensively distinguish truly important segments from various forms of redundant content that meaninglessly contribute to reasoning accuracy.

In this work, we systematically identify important segments that directly contribute to correct answer prediction within long CoTs, and show that unimportant segments cluster into distinctive redundant patterns. Specifically, we utilize integrated gradient (IG) attribution (Sundararajan et al., 2017) to calculate each token's direct influence on improving correct answer prediction and aggregate token-level attributions at the segment level to obtain two metrics: (1) *Attribution strength* quantifies the overall magnitude of a segment's influence on the model's prediction by summing absolute IG values within each segment with length normalization. (2) *Attribution direction consistency* is defined as the ratio between the absolute sum of signed IG attributions and the sum of absolute IG attributions, captures how uniformly a segment contributes in one direction (either positively or negatively toward the correct answer). Extremely high consistency often reflects shallow or skewed reasoning, such as segments with uniformly positive token IGs that merely provide superficial clarification (see the penultimate segments in Fig. 1 left), or uniformly negative token IGs that severely distort the reasoning process toward incorrect conclusions. In contrast, moderate consistency indicates more reflective reasoning that mixes supportive and corrective attribution within a segment, which is more critical for problem solving.

We first verify significant redundancy in long CoTs using *attribution strength*, showing that 30~40% of segments accumulate over 80% of total attribution in both correct and incorrect CoTs (Fig. 1, right-bottom). Building on this insight, we propose a segment-level selective learning framework that learns the most critical parts of long CoTs leveraging *attribution strength* and *direction consistency*. It identifies segments with high strength but moderate consistency as important, filtering out redundancies like repeated content, truncated thoughts, and dispensable clarifications with minimal gains on correct answer probability. Unlike pruning-based SFT methods that compress CoT supervision but compromise accuracy, our framework applies selective SFT (Lin et al., 2024) that trains only on important segments while masking loss for unimportant ones. This acts as implicit regularization by

preventing overfitting to uninformative content. Experiments across multiple models and datasets, using both self-generated and reference long CoTs, show that our method outperforms full-CoT SFT by improving reasoning efficacy (up to 4.7%) while reducing output length (up to 18%). Notably, our important segment identification can be broadly applied to other contexts, such as emphasizing policy gradient updates on important content in reinforcement learning.

## 2 METHODOLOGY

### 2.1 PRELIMINARY

A long-form CoT typically comprises multiple segments, each focusing on distinct aspects such as detailed problem understanding, intermediate reasoning exploring different solutions, or multiple verification process. Moreover, it inevitably contains redundant content, including repeated or truncated thought, and excessive clarification of self-evident facts, which diminishes the overall quality and efficiency of the CoT. To facilitate segment-level importance analysis of long CoTs, we first partition each long CoT $T$ into multiple segments $T = \{S_1, S_2, ..., S_n\}$ using common transition keywords (e.g., "\n\nWait", "\n\nAlternatively" ) that naturally occur within reasoning traces, following (Lu et al., 2025). The complete list of segmentation keywords is in the Appendix C.2.

**Importance Definition**  We define a segment within the full reasoning trace as important if it contributes to the final answer prediction, either by guiding transition from incorrect to correct answers or by enhancing confidence of correct answers. An intuitive approach is to sequentially append segments and force answer generation when adding each new segment to compute the change in correct answer probability as its attribution. However, this method underestimates segments that contribute indirectly, such as problem understanding or exploration of incorrect alternatives, which may not immediately improve answer prediction but establish crucial foundations for subsequent reasoning steps. Leave-one-out methods that mask individual segments and measure their effect on the final prediction, suffer from the same limitation, as omitting these segments typically does not significantly alter final answer prediction when their subsequent content remains intact. Therefore, we adopt a more principled approach using integrated gradients (IG) (Sundararajan et al., 2017) to measure each segment's attribution to the final answer prediction.

### 2.2 INTEGRATED GRADIENTS BASED SEGMENT IMPORTANCE

IG measures input token attribution by computing partial derivatives of the model output with respect to each input feature along a straight-line interpolation path from an uninformative baseline to the actual input embedding. By accumulating gradients across all interpolated points, IG estimates the total attribution of each input token to the final prediction. This approach captures both direct and indirect influences, making it particularly suitable for identifying the importance of reasoning segments that may contribute implicitly rather than immediately affecting the final answer prediction.

Formally, given a model $F$, an input token embedding $x$ and its baseline value $x'$ (typically the padding token embedding), the integrated gradient for the $i$-th input dimension is formulated as

$$\text{IG}_i(x) = (x_i - x_i') \times \int_{\alpha=0}^{1} \frac{\partial F(x' + \alpha \cdot (x - x'))}{\partial x_i} d\alpha \tag{1}$$

$$\approx (x_i - x_i') \times \frac{1}{J} \sum_{j=1}^{J} \frac{\partial F(x' + j/J \cdot (x - x'))}{\partial x_i} \tag{2}$$

where $x_i$ and $x_i'$ refer to the $i$-th dimension of $x$ and $x'$, respectively. The integral is approximated using J interpolation steps where $j$ denotes the $j$-th step.

**Segment-level Aggregation**  After computing IG attribution of each input token as $\text{IG}(x) = \sum_i \text{IG}_i(x)$, we aggregate them at the segment level to obtain segment-wise importance scores. While IG values can be either positive or negative, where positive values indicate increased likelihood of predicting the correct answer and negative values indicate a decrease, tokens with negative IG values should not be dismissed as unimportant. Negatively attributed tokens may represent incorrect but necessary exploratory reasoning that ultimately guides the model toward the correct solution. For

example, initial segments often exhibit overall negative IG sums, but these segments are typically important for establishing problem understanding and initial exploration. Therefore, we utilize the absolute IG value of each token to capturing the magnitude of influence regardless of direction.

We aggregate the absolute IG values for all tokens within each segment and define two segment-level measures: (1) Attribution Strength: computed as the sum of absolute IG values within the segment with square root length normalization applied to prevent bias toward longer segments; (2) Attribution Direction Consistency: measures the extent to which a segment exhibits consistent positive or negative contributions versus internally conflicting mixed attributions.

Formally, given a segment $S = \{o_1, \ldots, o_N\}$ with $N$ tokens, where each token $o_n$ is associated with an IG value $\text{IG}(o_n)$, we define the segment-level attribution strength and direction consistency as:

$$\text{Strength}(S) = \frac{\sum_{o_n \in S} |\text{IG}(o_n)|}{\sqrt{N}}, \quad \text{Consistency}(S) = \frac{\left| \sum_{o_n \in S} \text{IG}(o_n) \right|}{\sum_{o_n \in S} |\text{IG}(o_n)|}. \tag{3}$$

To ensure strength comparability across all segments within a CoT, we further normalize all strength scores across the set of segments $\{S_1, S_2, \ldots, S_M\}$. Specifically, for each segment $S_m$, the normalized attribution strength is defined as

$$\text{Strength}'(S_m) = \frac{\text{Strength}(S_m)}{\sum_{j=1}^{M} \text{Strength}(S_j)}. \tag{4}$$

## 2.3 SEGMENT-LEVEL SELECTIVE LEARNING

**Important Segment Identification** We utilize our segment-level metrics to distinguish critical segments from unimportant ones for more targeted learning of long CoTs. Important segments should exhibit higher attribution strengths that occupy a substantial portion of total attribution across all segments, and moderate rather than extremely high attribution direction consistency. Extremely consistent attribution directions (most positive or most negative) may indicate shallow or skewed reasoning (e.g., dispensable explanation of final answers, and uninformative or severe wrong explorations, as analyzed in Appendix A) while moderate consistency often reflects reflective and critical reasoning patterns where the model explores different possibilities, refines its understanding, and eliminates incorrect content, which are more valuable for learning effective reasoning strategies. Specifically, given a CoT with segments $\{S_1, S_2, \ldots, S_M\}$, we first rank segments in descending order of their normalized attribution strength. Let $\pi$ be a permutation such that:

$$\text{Strength}'(S_{\pi(1)}) \geq \text{Strength}'(S_{\pi(2)}) \geq \ldots \geq \text{Strength}'(S_{\pi(M)}) \tag{5}$$

We then identify the minimal number of top-ranked segments $k^*$ whose cumulative attribution exceeds a predefined threshold $\tau$ (e.g., 80%):

$$k^* = \arg \min_{k \in 1,2,\ldots,M} \left\{ \sum_{i=1}^{k} \text{Strength}'(S_{\pi(i)}) \geq \tau \right\} \tag{6}$$

Finally, we define the important segment set as those among the top-$k^*$ segments whose attribution direction consistency is below a threshold $\beta$ (e.g., 0.8), while the remaining as unimportant ones.

$$\mathcal{S}_{\text{important}} = \left\{ S_{\pi(i)} \mid i \leq k^*, \, \text{Consistency}(S_{\pi(i)}) \leq \beta \right\} \tag{7}$$

**Selective SFT** Long-CoT trajectories serve as valuable supervisory resources for cold-start of reasoning models via SFT, which optimizes parameters $\theta$ using the cross-entropy loss over all tokens.

$$L_{SFT}(\theta) = -\frac{1}{T} \sum_{t=1}^{T} \log P(o_t | o_{<t}, q; \theta) \tag{8}$$

where $T$ denotes the length of the reasoning trajectory, $o_t$ is the predicted token at position $t$, $o_{<t} = \{o_1, o_2, \ldots, o_{t-1}\}$ represents the preceding token sequence, and $q$ is the input query.

To mitigate learning from meaningless parts, prior efficiency works mainly prune redundant portions of each complete CoT, yielding more concise supervision, but pruning typically degrades performance. Instead, we follow (Lin et al., 2024) and propose to selectively train only on tokens within important

segments. This strategy ensures that parameter updates are driven solely by the most critical reasoning parts, while preserving coherence of the full trajectory. The selective SFT loss is formulated as:

$$L_{\text{Selective-SFT}}(\theta) = -\frac{1}{\sum_t I(o_t)} \sum_{t=1}^{T} I(o_t) \log P(o_t|o_{<t}, q; \theta) \tag{9}$$

where $I(o_t)$ indicating whether token $o_t$ belongs to a segment $S_m$ with $S_m \in \mathcal{S}_{\text{important}}$.

## 3  SEGMENT IMPORTANCE ANALYSIS

Before applying our segment-level selective learning strategy, we first validate that our importance metric effectively distinguishes meaningful reasoning parts from redundant behaviors and determine the optimal hyperparameters $\tau$ and $\beta$.

**Analysis Setup** We conduct an investigation on self-generated long reasoning trajectories from the LIMO datasets (Ye et al., 2025) using R1-Distill-Qwen2.5-7B (Guo et al., 2025). From 32 candidate samples per problem, we select the shortest correct and incorrect outputsm, split them into segments and compute their attribution strength and consistency. Segments are ranked as Eq. 5, with cumulative attribution averaged across percentage intervals, as illustrated in the top-right part of Fig. 1. The analysis reveals that only 30%~40% of segments contribute significantly (80%) to the final prediction, regardless of correctness, while most segments exhibit low attribution. This verifies substantial redundancy in long CoTs and motivates our approach for identifying important segments.

### 3.1  IMPORTANT VERSUS UNIMPORTANT SEGMENTS

We aggregate important and unimportant segments, respectively from all correct reasoning outputs[2] and compare the distinct patterns exhibited by these two subsets. We first intuitively set the threshold $\tau = 80\%, \beta = 0.95$ as Eq. 6, 7 for this comparative analysis.

**1. Important segments yield greater improvement in correct answer predictions.** We sequentially append segments to the input and enforce answer generation after adding each new segment to compute the change $\Delta$ in correct answer confidence relative to without that segment. The model's confidence on the correct answer is defined as the probability that it generates the correct answer. We empirically estimate this correct answer confidence by calculating the fraction of multiple temperature-sampled generations (i.e., 32 samples) that produce the correct answer. As shown in Fig. 2, segments with higher attribution strength (top 80% of total attribution), and moderate direction consistency achieve significantly larger gains in correct answer confidence compared to their counterparts. This suggests that segments with high strength but

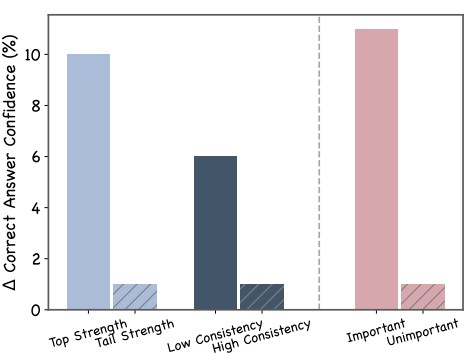

Figure 2: Change ($\Delta$) in correct answer confidence across different segment types.

moderate consistency reflect the critical exploratory reasoning that contribute to accurate predictions.

**2. Important segments exhibit relatively lower perplexity and entropy.** We calculate the log perplexity and entropy of all tokens in each long reasoning trace and aggregate them at the segment level to compare the linguistic properties between important and unimportant segments. As shown in Fig. 3, important segments consistently exhibit significantly lower log perplexity and entropy compared to unimportant segments. This indicates that our identified critical segments tend to be more predictable and carry lower uncertainty, likely reflecting processes such as problem understanding, logical deduction or essential computations that follow more constrained patterns. In contrast, unimportant segments, such as incomplete truncations associated with higher uncertainty (Wang et al., 2025d), as well as verbose explanations or dispensable elaborations that exhibit higher variability and greater linguistic freedom, resulting in higher perplexity and entropy.

---

[2]We focus on correct reasoning outputs, as our objective is to leverage reliable reasoning traces for SFT.

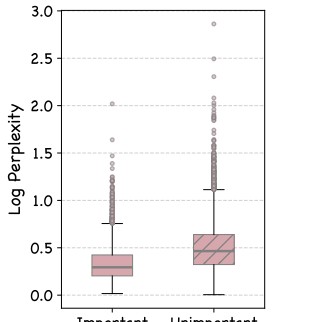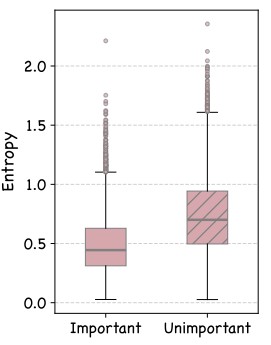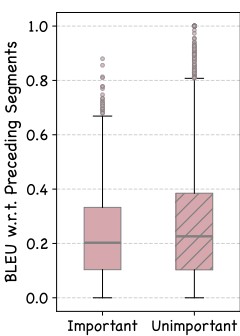

Figure 3: Log perplexity, entropy and BLEU similarity of important versus unimportant segments.

**3. Unimportant segments are more characterized by repetition or incomplete truncation.** We further examine whether unimportant segments are closely associated with redundant reasoning behaviors such as repetition and incomplete truncation. To this end, we calculate BLUE similarity of each segment against all preceding segments within the same CoT (rightmost panel, Fig.3). Unimportant segments exhibit overall higher BLEU similarity scores compared to important segments. In particular, unimportant segments contain substantially more highly repetitive content (e.g., BLEU $> 0.8$) that reiterate previously established reasoning. In addition, we prompt Qwen3-8B (Yang et al., 2025) to assess whether each segment is incompletely truncated, meaning that subsequent segments failing to logically follow it (as implemented in Appendix B). The results show that 49% of unimportant segments are classified as truncated compared with only 26% of important segments. These results demonstrate that unimportant segments identified by our method are more strongly associated with repetition and truncation, contributing minimal additional value to the reasoning chain and can even introduce noise that obscures the logical flow. We additionally analyze the positional distribution of important versus unimportant segments within each CoT in Appendix A.

## 3.2 HYPERPARAMETER SEARCH

To determine the optimal threshold $\tau$ for identifying high-impact segments for targeted training, we adopt a greedy search strategy by selecting $\tau^*$ that maximizes the difference in correct answer confidence change $\Delta$ between resulted important and unimportant segments while minimizing false negatives (i.e., ensuring that unimportant segments exhibit the lowest confidence gain). As shown in Fig. 4, we set the optimal threshold as $\tau = 0.7$. On average, this setting yields about 33% of segments classified as important, accounting for 45% of the tokens within each self-generated CoT. This imbalance is because important segments tend to be longer, whereas unimportant ones often either repeat part of the previous content or are prematurely truncated in the middle. We further ablate different values of $\beta$ under $\tau = 0.7$ and provide experimental comparison of varying $\tau$ on overall performance in Appendix D and identify $\beta = 0.8$ as the optimal consistency threshold.

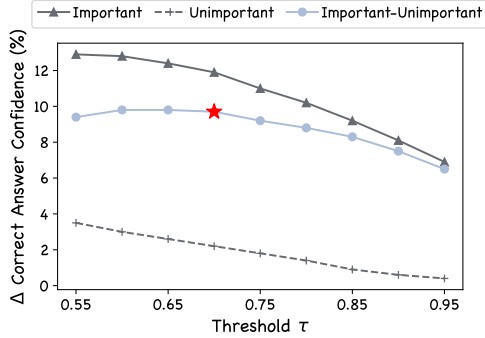

Figure 4: Change in correct answer confidence ($\Delta$) across different segment types under varying $\tau$. The red star marks the selected threshold $\tau^*$.

## 4 EXPERIMENTS

### 4.1 SETUP

**Training Details** We compare our selective SFT on IG-based important segments against standard SFT on full long CoT supervision, experimenting both instruction-following and reasoning baseline

Table 1: Comparison results on different baseline models. "Overall" reports the average accuracy under greedy decoding, or pass@1 and pass@6 under temperature sampling, together with the average output token count over all benchmarks. The percentages in parentheses indicate the relative accuracy improvement and token reduction of our method compared to full CoT SFT during generation. The best results are highlighted in **bold**.

| Models | In Domain | | | Out of Domain | | | Overall | | |
|---|---|---|---|---|---|---|---|---|---|
| | MATH500 | AMC23 | AIME24 | GPQA | Minerva | Olympiad | Acc./Pass@1 | Pass@6 | Length |
| *Greedy Decoding* | | | | | | | | | |
| R1-Distill-Qwen-1.5B | 70.6 | 52.5 | 16.7 | 27.3 | 21.7 | 31.4 | 36.7 | / | 20244 |
| Full CoT SFT | 80.8 | **70.0** | 26.7 | 25.3 | 26.8 | 39.4 | 44.8 | / | 16520 |
| **Segment Selective SFT** | **82.4** | 60.0 | **40.0** | **28.8** | **27.6** | **42.8** | **46.9** (↑4.7%) | / | 13506 (↓18.2%) |
| R1-Distill-Qwen-7B | 85.8 | 85.0 | 46.7 | **42.9** | 38.2 | 47.9 | 57.7 | / | 12518 |
| Full CoT SFT | 91.2 | 90.0 | 50.0 | 42.4 | 41.2 | 57.9 | 62.1 | / | 9693 |
| **Segment Selective SFT** | **95.2** | **90.0** | **56.7** | 40.4 | **46.3** | **58.7** | **64.5** (↑3.9%) | / | 8499 (↓12.3%) |
| Qwen2.5-7B-Instruct | 77.0 | 50.0 | 10.0 | 38.9 | **34.9** | 37.5 | 41.4 | / | 1405 |
| Full CoT SFT | **77.2** | 55.0 | 23.3 | 38.9 | 29.8 | 40.9 | 44.2 | / | 10317 |
| **Segment Selective SFT** | 76.6 | **57.5** | **23.3** | **44.4** | 30.5 | **41.5** | **45.6** (↑3.2%) | / | 9852 (↓4.5%) |
| *Temperature Sampling* | | | | | | | | | |
| R1-Distill-Qwen-1.5B | 84.0 | 73.3 | 29.8 | 32.8 | **32.0** | 44.8 | 49.5 | 71.1 | 9791 |
| Full CoT SFT | 85.1 | 74.7 | 35.1 | 31.0 | 32.0 | 47.3 | 50.9 | 72.4 | 10043 |
| **Segment Selective SFT** | **85.1** | **75.9** | **36.1** | **32.8** | 31.9 | **48.1** | **51.7** (↑1.6%) | **73.2** (↑1.1%) | 9388 (↓6.5%) |
| R1-Distill-Qwen-7B | 92.8 | 90.5 | 54.2 | **46.3** | 45.5 | 57.7 | 64.5 | 79.2 | 7593 |
| Full CoT SFT | **93.6** | 91.5 | 60.2 | 41.8 | 45.6 | 60.3 | 65.5 | 79.2 | 7934 |
| **Segment Selective SFT** | 93.3 | **91.9** | **61.0** | 42.3 | **45.8** | 60.3 | **65.8** (↑0.5%) | **80.0** (↑1.0%) | 7709 (↓2.8%) |
| Qwen2.5-7B-Instruct | 75.7 | 51.9 | 12.0 | 36.9 | **35.4** | 37.2 | 41.5 | 60.2 | 910 |
| Full CoT SFT | 77.0 | 55.4 | 16.4 | 38.9 | 31.6 | 40.8 | 43.4 | 66.0 | 9902 |
| **Segment Selective SFT** | **77.1** | **58.5** | **16.4** | **43.7** | 32.4 | **41.9** | **45.0** (↑3.7%) | **66.5** (↑0.8%) | 9195 (↓7.1%) |

models of different scales: R1-Distill-Qwen2.5-1.5B, R1-Distill-Qwen2.5-7B (Guo et al., 2025) and Qwen2.5-7B-Instruct (Team, 2024). We use 817 high-quality questions from the LIMO mathematical dataset (Ye et al., 2025) and evaluate both provided and self-generated CoT supervision. For self-generated supervision, we prompt R1-Distill-Qwen2.5-7B to generate 32 candidate responses per question and select the shortest correct response for training (as described in Sec. 3). We utilize provided higher-quality CoT supervision for R1-Distill-Qwen2.5-1.5B and R1-Distill-Qwen2.5-7B, and self-generated CoTs for Qwen2.5-7B-Instruct. We apply the same hyperparameter configurations of $\tau = 0.7$ and $\beta = 0.8$ to both sources of CoT supervision (as their hyperparameter search in Sec. 3.2 shows similar results). All models are fine-tuned with full parameters with a maximum sequence length of 16384. Additional training details are provided in Appendix C.3. We also provide model performance under different combination of $\tau$ and $\beta$ in Appendix D.

**Evaluation Details** We conduct evaluation on both in-domain and out-of-domain benchmarks following the similar setup in (Ye et al., 2025). The in-domain benchmarks include Math500 (Hendrycks et al., 2021), AMC23 and AIME24 while GPQA-Diamond (Rein et al., 2024), Minerva (Lewkowycz et al., 2022) and OlympiadBench (He et al., 2024) are out-of-domain. We report accuracy using greedy decoding, and pass@1 and pass@6 metrics using temperature sampling (temperature=0.6, top-p=1.0) as no universally optimal decoding strategy exists across all models and benchmarks. We employ a zero-shot CoT setting with a maximum response length of 32,768 tokens. For larger benchmarks (MATH500, GPQA-Diamond, Minerva, and OlympiadBench), we sample 6 responses per instance, while for smaller benchmarks (AMC23 and AIME24), we sample 32 responses per instance. We compute the average token count across all sampled outputs as the response length.

## 4.2 MAIN RESULTS

Table 1 reports the overall comparison across six benchmarks. Relative to the baseline performance without SFT training, our segment-level selective SFT consistently outperforms full CoT SFT on both in-domain and out-of-domain datasets, across different baseline models and under both greedy decoding and temperature sampling. In addition, compared to full CoT SFT, our approach achieves a substantial reduction in token usage, leading to better optimization while simultaneously improving output efficiency. Notably, the gains in both accuracy and token reduction are more pronounced under greedy decoding. This is because the randomness in multiple temperature sampling tends to

smooth out and partially offset the advantages of training. Nevertheless, even under such stochastic conditions, our approach still yields measurable improvements, underscoring its robust effectiveness.

## 4.3 ABLATION STUDY

To investigate the contributions of our important segment identification and selective SFT, we conduct an ablation study using R1-Distill-Qwen-1.5B with results averaged across datasets. Taking full CoT SFT as the baseline, we compare two categories *Pruned CoT SFT* according to our identified important segments and *Selective SFT* using various criteria: (1) randomly selecting roughly 33% of segments from each CoT; (2) selecting the top 45% of tokens ranked by absolute IG values; (3) Selecting the top 45% of tokens ranked by original IG values; (4) Selecting only high-strength segments; (5) selecting segment with high strength and moderate direction consistency (our method). These ratios are chosen to ensure the amount of selected content is comparable across methods.

As shown in Table 2, pruning unimportant segments degrades accuracy, whereas our selective learning can improve SFT performance while reducing token usage. Among selective SFT, our IG-based important segments outperforms random segments and token-level selection based on absolute or original IG values, with absolute IG values proving more effective than original IGs for identifying importance. Furthermore, the improvement over only high-strength segments shows that incorporating moderate direction consistency can further filter out low-impacted segments to achieve greater token reduction without accuracy loss. Overall, these consistent improvements demonstrate that segment-level granularity better preserves reasoning coherence than token-level selection, and our IG-based strength-consistency criterion effectively identifies the most contributive reasoning components.

Table 2: Ablation study on R1-Distill-Qwen-1.5B. Our important segments have high attribution strength and moderate direction consistency.

| Methods | Greedy | | Sampling | |
|---|---|---|---|---|
| | Acc. | Length | Pass@1 | Length |
| Full CoT SFT | 44.8 | 16520 | 50.9 | 10043 |
| Pruned CoT SFT | 43.9 | 15097 | 49.2 | 7766 |
| Selective SFT | | | | |
| Random Segments | 45.1 | 15138 | 50.8 | 10117 |
| High-Abs-IG Tokens | 46.1 | 14612 | 50.8 | 9495 |
| High-Orig.-IG Tokens | 45.2 | 14747 | 50.3 | 9876 |
| High Strength Segments | 46.9 | 14715 | 51.4 | 9466 |
| Our Important Segments | 46.9 | 13506 | 51.7 | 9388 |

## 4.4 COMPARISON TO EXISTING SEGMENT IMPORTANCE MEASURES

To demonstrate the superiority of our IG-based importance segment identification, we apply Selective SFT using our identified segments and compare against alternative importance measures: (1) **First-Correct Solution** (Chen et al., 2024): retaining the first correct answer and all preceding segments as important; (2) **Confidence-Gain Segments** (Xu et al., 2025): identifying segments that directly improve the model's confidence in the correct answer compared to when the segment is removed (as described in Sec. 3.1). (3) **Segment Perplexity** (Cui et al., 2025b): identifying critical segments whose removal leads to a substantial increase in CoT perplexity. (4) **Segment Entropy** Li et al. (2025a): prioritizing segments with higher aggregated token-level entropy as they indicate greater informational contribution. We set thresholds for methods (3) and (4) to ensure the selected important content occupies approximately the same 45∼50% token ratio as our method for a fair comparison.

Results averaged across all datasets on R1-Distill-Qwen-1.5B are shown in Table 3. Our method consistently outperforms others in both accuracy and token reduction, demonstrating its ability to more precisely identify segments that are truly contributive to reasoning, while perplexity and entropy are not entirely consistent signals of importance. The weak performance of Confidence-Gain Segments stems from its neglect of indirectly contributive segments, such as those supporting early-stage question understanding or eliminating incorrect solution paths, as evident in its identification of only 24% of segments on average. Notably, our approach

Table 3: Selective SFT performance on important segments using different importance measures on R1-Distill-Qwen-1.5B. Accuracy and token length are averaged across all datasets.

| Methods | Greedy | | Sampling | |
|---|---|---|---|---|
| | Acc. | Length | Pass@1 | Length |
| Full CoT SFT | 44.8 | 16520 | 50.9 | 10043 |
| First-Correct Solution | 46.2 | 15238 | 51.0 | 9492 |
| Confidence-Gain Segments | 44.7 | 15479 | 50.3 | 9856 |
| Segment Perplexity | 44.7 | 14973 | 50.2 | 9816 |
| Segment Entropy | 44.5 | 16288 | 51.2 | 10148 |
| IG-based Important Segments | 46.9 | 13506 | 51.7 | 9388 |

surpasses First-Correct Solution, suggesting that segments beyond the first correct answer, such as answer verification or alternative solutions, are also meaningful by improving correct answer confidence. Moreover, it achieves greater token reduction than First-Correct Solution, indicating it can more selectively mask unnecessary segments before the first correct answer.

## 4.5 PRUNING-BASED COMPARISON

We further apply our IG-based importance segment identification to segment-level CoT pruning for SFT, and compare against other strategies introduced in Sec. 4.4. We do not compare with token-level pruning methods as they would significantly disrupt text coherence and fluency in long CoTs and severely degrades SFT performance. We adjust thresholds for our method and compared methods (3) and (4) to prune nearly 30% of tokens from unimportant segments, as excessive pruning ratios substantially impact SFT performance. As shown in Table 4, our method better maintains performance in both decoding settings, compared to other CoT pruning strategies. Among them, First-Correct Solution performs best because it retains consecutive and complete reasoning processes, while other methods inevitably affect the information coherence of supervisions, leading to more severe performance drops and limited token reduction in long-CoT scenarios. However, performance still declines after First-Correct Solution pruning. Compared to Table 3, these results further demonstrate the effectiveness of our proposed segment-level selective learning across various importance metrics, which can maintain or even improve performance while reducing token consumption.

Table 4: Performance of different segment-level pruning methods for CoT SFT on R1-Distill-Qwen-1.5B. Accuracy and token length are averaged across all datasets.

| Methods | Greedy | | Sampling | |
|---|---|---|---|---|
| | Acc. | Length | Pass@1 | Length |
| Full CoT SFT | 43.5 | 17140 | 50.7 | 10163 |
| First-Correct Solution | 45.2 | 11987 | 47.2 | 6846 |
| Confidence-Gain Segments | 40.3 | 11344 | 44.5 | 4180 |
| Segment Perplexity | 38.6 | 15892 | 47.9 | 8847 |
| Segment Entropy | 43.4 | 14321 | 45.9 | 7307 |
| IG-based Important Segments | 43.9 | 15097 | 49.2 | 7766 |

## 5 RELATED WORK

**LLMs Reasoning Efficiency** LRMs exhibit strong reasoning capabilities but typically generate verbose and redundant traces, leading to high computational costs and error accumulation (Sui et al., 2025; Wang et al., 2025a;d). Recent research explores various approaches to improve reasoning efficiency, which can be grouped into four categories. First, prompt-based methods design input prompts to control task difficulty and token budgets (Han et al., 2025; Renze & Guven, 2024). Second, decoding strategies dynamically reduce redundancy and shorten outputs during inference (Wang et al., 2025c; Liao et al., 2025). Third, latent reasoning approaches represent reasoning trajectories in latent spaces (Hao et al., 2024; Deng et al., 2024; Dong et al., 2025). Finally, training-based methods use SFT on compressed CoT supervision (Xia et al., 2025a; Lu et al., 2025) and reinforcement learning (RL) with conciseness rewards (Yuan et al., 2025; Lou et al., 2025) or encouraged exploration (Xing et al., 2025). We focus on SFT-based methods, as they better balance efficiency and performance while serving as essential cold starts for RL-based training.

**Importance Measurement** To construct compressed CoT supervision for SFT-based methods, a key direction is to measure importance of different parts within full CoTs and prune less important content. Token-level methods typically utilize token logits and entropy to prioritize salient tokens but ignore semantic coherence (Xia et al., 2025a; Cheng et al., 2025; Wang et al., 2025b) Segment-level methods assess importance at segment granularity, better aligning with human reasoning units (Li et al., 2025a; Cui et al., 2025b). Moreover, Xu et al. (2025) propose leave-one-out and greedy forward selection to estimate segment attribution. Recent Wang et al. (2025e) suggests that LRMs rely on both reasoning and memory retrieval mechanisms, and introduces perturbation-based attribution to promote reasoning-dominant behavior. Unlike these indirect measures, we use integrated gradients–based attribution method to quantify the direct contribution of each segments to final answer predictions.

**Selective Training** Recent research challenges traditional full sequence learning paradigm that uniformly optimizes loss on all tokens (Lai et al., 2024; Lin et al., 2025). Lin et al. (2024) selectively apply loss only to useful tokens and improve pretraining performance. Hans et al. (2024) propose training on randomly selected token prevents memorization without performance degradation. Kim et al. (2025) groups tokens in each sample by importance and optimizes weighted loss that adap-

tively emphasizes challenging groups. In this work, we propose a segment-level selective learning framework that masks unimportant segments labeled by our importance measurement during SFT.

## 6 CONCLUSION

In this work, we proposed a segment-level selective learning framework for more effective learning from long reasoning traces. By incorporating integrated gradient attribution, we introduced two segment-level metrics: *attribution strength* and *attribution direction consistency* to identify important segments with high strength but moderate consistency from full CoTs. Selective SFT on these segments improves both reasoning accuracy and reduces output length, outperforming full-CoT SFT. Our approach provides a principled way to identify critical reasoning segments and offers broader potential for emphasizing policy updates on targeted content in reinforcement learning.

## REPRODUCIBILITY STATEMENT

To support reproducibility, we provide detailed descriptions of our metrics and framework in Sec. 2 and Appendix C.4. Additionally, we include comprehensive implementation details to reproduce our results in Sec. 4.1 and Appendix. C.3, covering hyperparameters selection, training and evaluation details, training frameworks. We will release our data and code after the anonymous review process.

## ACKNOWLEDGMENTS

This research is supported in part by the Office of the Director of National Intelligence (ODNI), Intelligence Advanced Research Projects Activity (IARPA), via the HIATUS Program contract #2022-22072200006, the Defense Advanced Research Projects Agency with award HR00112220046, and NSF IIS 2048211. We would like to thank all the collaborators in USC INK research lab for their constructive feedback on the work.

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

## A  POSITIONAL DISTRIBUTIONAL OF SEGMENTS IMPORTANCE

We analyze the positional distribution of segment importance within long CoTs by identifying the decision segment, defined as the first segment that derives the correct answer. We observe 40% of important segments appear after the decision segment, indicating that additional verification or exploration beyond the initial correct answer can still play a crucial role. In contrast, 57% of unimportant segments occur before the decision segment, suggesting that the search for the correct answer often involves redundant behaviors such as repetition and truncation. This also highlights that simply splitting by decision segments may introduce a considerable number of false positives and false negatives (Wang et al., 2025d). A finer-grained analysis further reveals that most low-strength unimportant segments (64%) occur before the decision segment. In contrast, the majority of high-consistency unimportant segments (72%) appear after it. Combined with Figure 3.1 shows that high-consistency segments typically induce minimal improvement in correct-answer confidence, this suggests that once the correct answer is found, such segments are prone to shallow or redundant elaboration as formalized verification or reinforcement of the already-found answer rather than introducing deeper critical examination.

## B  INCOMPLETELY TRUNCATED SEGMENT IDENTIFICATION

To determine whether a given segment is complete or truncated by alternative content, we employ Qwen3-8B with a carefully designed prompt template. The template, shown in Table 5, describes the task and includes few-shot examples. It instructs the model to first reason step by step in thinking mode, and then output a final judgment on whether Segment 2, given the surrounding context of Segment 1 and Segment 3, is a truncated segment. We frame this as a 0/1 classification task rather than using Yes/No answers, since we find that restricting the output to binary digits yields more reliable performance.

For each segment, we conduct up to two inference rounds. In round 1, the model runs in thinking mode with a token budget of 2000, temperature 0.2, top_p 0.7, and repetition_penalty 1.1. If the model fails to reach to a conclusion in this round, we will inject an **Early-Stopping Prompt** (as shown in Table 6), and run the model in round 2, with temperature 0.1, top_p 0.7, max_tokens 1000, and repetition_penalty 1.1. The input for round 2 consists of the original input and the Round 1 output, followed by the injected Early-Stopping Prompt.

## C  SUPPLEMENTARY IMPLEMENTATION DETAILS

### C.1  LLM USAGE

Our use of LLMs in this paper comprises two main components: writing assistance and tools for data analysis and construction. During writing, we utilize ChatGPT 5 and Claude Sonnet 4 to help polish the exposition. For data analysis and construction, as described in Section 4.1, we employ R1-Distill-Qwen2.5-7B to generate its CoT trajectories leading to correct answers as supervision for SFT, and to compute integrated gradient attributions on each token.

### C.2  SEGMENTATION KEYWORDS

To partition each long CoT into individual segments, we select transition keywords following (Lu et al., 2025) which includes "But", "Wait", "Alternatively", "However", "Hmm", "Hmmm", "Not sure", "Going back", "Backtrack", "Trace back", and "Another". Specifically, most of our keywords are directly adopted from (Lu et al., 2025), including "\n\nWait", "\n\nAlternatively", "\n\nHowever", "\n\nNot sure", "\n\nGoing back", "\n\nBacktrack", "\n\nTrace back", "\n\nAnother". We prepend "\n\n" to each keyword because many LRMs naturally structure their reasoning output with paragraph breaks, making this prefix a reliable delimiter for identifying reasoning segments. Our keywords differ from Lu et al. (2025) in two ways. First, instead of the broad keyword "But", we use more specific variants including "\n\nBut wait", "\n\nBut alternatively" and "\n\nBut just to". This change is motivated by empirical observations that LRMs, especially smaller distilled models like R1-Distill-Qwen2.5-7B, tend to use "But" as conversational fillers rather than a meaningful reasoning transition. Second, we exclude fillers such as "Hmm" and "Hmmm", which we also find to be frequently

---

**Prompt for Judging Truncated Segments**

I will provide you with 3 segments of text which are generated continuously by a reasoning model. Each segment itself represents a complete thinking step, and the 3 segments together represent 3 continuous thinking steps during the reasoning process.

Your task:
Taking Segment 1 and Segment 3 as context, determine whether Segment 2 is a "truncated" segment, i.e., an unfinished thinking step.

Example of a non-truncated segment:
> But let me double-check my calculations to be sure I didn't make any mistakes.\n\nFirst, the original equation:\n\n$\sqrt{}$(1995) * x^(log_1995 x) = x^2.\n\nI took log base 1995 of both sides, correctly applied the logarithm rules, and ended up with a quadratic in y = log_1995 x. Solving that quadratic gave me two solutions, which translated back to x gave me two roots. Multiplying them gave me 1995², which is 3,980,025. So, yes, the last three digits are 025.

The above segment shows a complete thought process, because: 1. Internally, this segment has a clear logic, where calculations and reasoning are derived progressively. 2. It reaches a conclusion, which is the final answer to this reasoning step. 3. Externally, if given more context, this segment may continue or refute it's previous segment, and the next segment may also continue reasoning based on this segment, which forms a coherent reasoning chain.

Example of a truncated segment:
> Wait, hold on a second. Let me verify if 1995² is indeed 3,980,025.\n\nCalculating 1995 * 1995:\n\nI can compute 2000 * 2000 = 4,000,000.\n\nThen, 2000 * (-5) = -10,000.\n\nSimilarly, (-5) * 2000 = -10,000.\n\nAnd (-5) * (-5) = 25.

The above segment is a truncated segment, because: 1. Internally, this segment lacks a clear logic, and it only shows partial calculations without reaching the final conclusion of 1995². 2. Externally, if given more context, this segment may abandon it's previous segment, and the next segment may also abandon this segment, which breaks the reasoning chain.

Now, based on the definitions and examples above, please judge whether Segment 2 is a truncated segment. Here are the given 3 segments:
Segment 1:
> {SEGMENT 1}
Segment 2:
> {SEGMENT 2}
Segment 3:
> {SEGMENT 3}

Please reason step by step, and answer "1" or "0" in the following format:

My final answer is:\n\n$$\n\boxed{1 or 0}\n$$

NOTE:
- The final answer must be either "1" or "0", which means yes or no that Segment 2 is a truncated segment.
- The final answer must be in \boxed{} format.
- You must not explain anything more after giving the final answer.

Table 5: The prompt for judging truncated segments using Qwen3-8B.

---

**Early-Stopping Prompt**

\n\n Considering the limited time by the user, I have to immediately stop reasoning and give the answer (1 or 0) directly now.\n</think>\n\n My final answer is:\n\n

Table 6: Early-stopping prompt for terminating the model's thinking process.

used as verbal habits instead of genuine segment boundaries. These design choices aim to balance generalizability across LRMs while ensuring that segmented reasoning units are meaningful rather than overly fragmented.

To illustrate the effectiveness of our method using different keywords, we also experiment with using the exact keyword list from (Lu et al., 2025). We apply these keywords to segment the original LIMO CoTs and retrain R1-Distill-Qwen2.5-1.5B. This results in much finer-grained segmentation, increasing the average number of segments from 22.5 to 33.5 per reasoning trace. The selective SFT results in Table 7 show that this finer segmentation leads to higher accuracy and more concise outputs. We attribute this to that LIMO traces are generated by higher-quality models (e.g., DeepSeek-R1, QwQ-32B), whose reasoning contains fewer meaningless fillers. In such cases, finer segmentation yields more precise identification of important reasoning units.

Table 7: Results on R1-Distill-Qwen-1.5B using new transition keywords from (Lu et al., 2025).

| Methods | Greedy | | Sampling | |
|---|---|---|---|---|
| | Acc. | Length | Pass@1 | Length |
| R1-Distill-Qwen-1.5B | 36.7 | 20244 | 49.5 | 9791 |
| Full CoT SFT | 44.8 | 16520 | 50.9 | 10043 |
| Segment Selective SFT (Original Keywords) | 46.9 | 13506 | 51.7 | 9388 |
| Segment Selective SFT (New Keywords) | 51.5 | 12646 | 51.5 | 9167 |

### C.3 TRAINING & EVALUATION DETAILS

All models are trained using the Unsloth training framework (Daniel Han & team, 2023). Specifically, we train R1-Distill-Qwen2.5-1.5B and R1-Distill-Qwen2.5-7B for 10 epochs respectively with a learning rate of 3e-5 and 1.5e-5. For Qwen2.5-7B-Instruct which does not original support long-chain reasoning, we adopt a two-stage SFT strategy that first conducts full CoT training for 7 epochs with a learning rate of 1.5e-5, and continued training for 4 additional epochs using only the identified important segments, with a learning rate of 8e-6. For a fair comparison, the full CoT SFT baseline is trained with the same two-stage schedule. All experiments employ a cosine learning rate schedule. The overall performance (including greedy accuracy, sampling pass@1, pass@6 and output length) reported in this paper is calculated using macro averaging across all datasets, treating each dataset as equally important.

### C.4 METHODOLOGY DETAILS

For calculating token-level integrated gradients, we use $J = 50$ interpolation steps to estimate the integral in the IG computation for the cost-precision trade-off as indicated by (Sundararajan et al., 2017) that 20∼300 integration steps typically approximate the path integral within about 5% approximation error. We specifically utilize R1-Distill-Qwen2.5-7B to calculate token IG values on both reference long CoTs provided by the LIMO dataset and self-generated long CoT supervisions by Distill-Qwen2.5-7B. Each costs roughly 7 (GPU×hours), compared to the 8 (GPU×hours) required for a full SFT run of the same 7B model. In deploying our method and all baselines, we additionally include the first and last segments which are responsible for establishing the problem understanding and for explicitly formatting the final answer, together with the identified important segments as learning objectives in both selective SFT and pruning-based settings. For segment-level aggregation, we have explored several strategies: (1) length-normalized summation, (2) direct summation, and (3) averaging the top 20% of token IG values. We find that the first two achieve comparable performance, while length-normalized summation better mitigates bias toward longer segments.

## D DIFFERENT THRESHOLD HYPERPARAMETERS

We further experiment with different selection of the hyperparameters $\tau$ and $beta$ on R1-Distill-Qwen2.5-1.5B. We first explore different values of $\beta \in \{0.7, 0.8, 0.9\}$ under a fixed $\tau = 0.7$ and report the temperature sampling results in the left panel of Figure 5. We observe that $\beta = 0.8$ achieves the best performance, which is also adopted as the main experimental setting in our paper. Based on $\beta = 0.8$, we further validate whether $\tau = 0.7$ indeed achieves the best experimental performance as expected in our hyperparameter search. We vary $\tau \in \{0.6, 0.7, 0.8, 0.9\}$ as shown in the right panel

of Figure 5. Results show that as $\tau$ increases, more segments are selected as important among all candidates. Consequently, the token reduction after training becomes smaller. However, performance decreases because higher $\tau$ introduces more false positives, and a denser loss mask negatively impacts training. Overall, $\tau = 0.7$ and $\beta = 0.8$ constitute our final experimental setting.

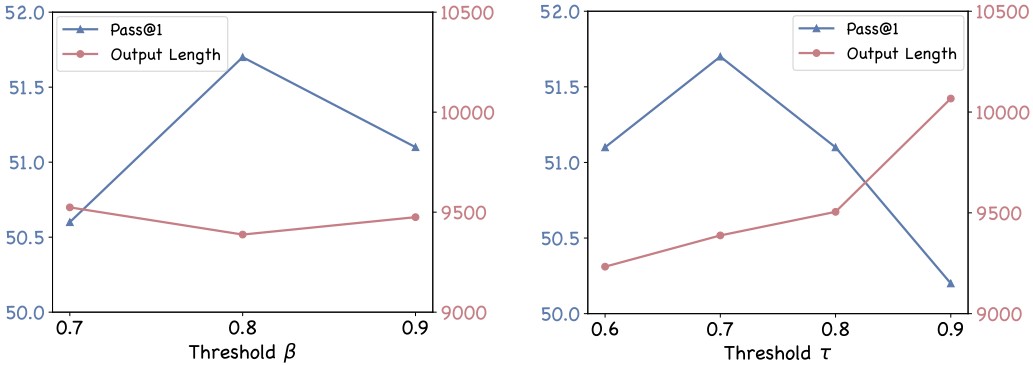

Figure 5: Model performance under different hyperparameters $\tau$ and $\beta$.

## E  ADDITIONAL RESULTS ON LLAMA3.1-8B-INSTRUCT.

To further demonstrate the effectiveness of our method beyond the Qwen model family, we additionally train LLama3.1-8B-Instruct using the LIMO dataset. As shown in Table 8, Segment Selective SFT still achieves performance improvement, demonstrating that our method generalizes well to a different model family.

Table 8: Comparison results on LLaMA3.1-8B-Instruct.

| Methods | Greedy | | Sampling | |
|---|---|---|---|---|
| | Acc. | Length | Pass@1 | Length |
| LLaMA3.1-8B-Instruct | 24.0 | 9691 | 23.5 | 4161 |
| Full CoT SFT | 30.4 | 17474 | 33.0 | 15067 |
| Segment Selective SFT | 33.8 | 16005 | 33.5 | 14669 |

