# OpenReview forum: "Segment-Level Attribution for Selective Learning of Long Reasoning Traces"
_ICLR.cc/2026/Conference — ICLR 2026 Poster_

### Official Review · Reviewer_sHrL · 2025-10-29

**Soundness:** 3
**Presentation:** 3
**Contribution:** 3
**Rating:** 6
**Confidence:** 4

**Summary:**

This paper study on the segment selection in long chains of thought (CoT), they propose a series of new metrics: attribution strength and direction consistency, from integrated gradient attribution perspective. Based on these two metrics, this paper proposes a segment-level selective learning framework to identify important segments with high attribution strength but reflective than shallow reasoning. The framework then applies selective SFT on these important segments while masking loss for unimportant ones. Experiments across multiple models and datasets show the approach improves accuracy and output efficiency, enabling more effective learning from long reasoning traces.

**Strengths:**

Compared to prior studies which either neglects semantic integrity and fails to interpret redundancy or ignore the indirect effect to the answer, this paper integrated gradient attribution to analyse segment's effect in depth.

**Weaknesses:**

The mechanism in section 2.2 "INTEGRATED GRADIENTS BASED SEGMENT IMPORTANCE" should be discussed more, see questions part for details.

**Questions:**

1. What is the definition of confidence in Fig. 2 and Fig. 4.
2. In equation (2), how exactly is IG be calculated from hidden state to outcomes, i.e. the partial derivative, how to define J in practice. If it includes frequently altering input to acquire output score, computation cost should be discussed.
3. Most of the experiment is conducted under Qwen-family framework, which leave the possibility that this method benefits from the model's characteristics cannot be completely ruled out. Experiment under different model architecture is suggested.
4. Is there more application scenarios could be employed by those IG metrics.

---

> ### Author Response · Authors · 2025-11-27
> **Response to Reviewer sHrL (1/2)**
>
> Thank you for your detailed comments and constructive feedback! Below are our responses to each of them.
>
> > Q1: What is the definition of confidence in Fig. 2 and Fig. 4.
>
> We apologize for the confusion. As described in lines 240–243, "We estimate the correct answer confidence by multiple temperature-sampled generations (i.e., 32 samples) to calculate the frequency of correct answers." In other words, the model’s confidence on the correct answer is defined as the probability that it generates the correct answer. However, directly computing this probability from the model’s token-level output distribution after each segment can be unreliable because (1) the same answer may have multiple valid surface forms (especially numeric answers), and (2) formatting tokens may affect the computed probability. To avoid these issues, we estimate confidence empirically as **the fraction of sampled generations that produce the correct answer**.
>
> ---
>
> > W & Q2: In equation (2), how exactly is IG be calculated from hidden state to outcomes, i.e. the partial derivative, how to define J in practice. If it includes frequently altering input to acquire output score, computation cost should be discussed.
> 1. **How IG is computed from hidden states to outcomes.** In Eq. (2), we compute the partial derivatives of the model output with respect to each input embedding. The scalar output $F(x)$ is defined as the negative cross-entropy loss of the answer tokens, equivalently the sum of log-likelihoods of the answer sequence. Specifically, at each interpolation point, we run a forward pass to obtain $F(x)$ and then backpropagate its gradient to the input embeddings.
> 2. **How $J$ is chosen in practice.** Prior work on IG shows that 20~300 integration steps typically approximate the path integral within about 5% approximation error [1,2]. Considering the cost-precision trade-off, we select $J=50$, which provides stable attributions while keeping computation tractable.
> 3. **Computation cost.** IG evaluates the model along a straight-line interpolation path from a baseline embedding to the actual input, yielding more faithful attributions than a single-step gradient but introducing extra cost proportional to $J$. Larger $J$ reduces approximation error but increases runtime. In our experiments, using R1-Distill-Qwen2.5-7B with $J=50$ costs roughly 7 (GPU$\times$hours), compared to the 8 (GPU$\times$hours) needed for a full SFT run of the same 7B model.
> 4. **Cost reduction.** The cost can be further reduced by decreasing the interpolation step $J$. Our additional experiments (in Table R4) show that using a smaller $J$ ($J=20$) still yields strong attribution quality with a minimal performance drop while still outperforming full CoT SFT. In this setting, IG computation becomes much more efficient, requiring only about 3 GPU$\times$hours.
>
> [1] Axiomatic Attribution for Deep Networks.
> [2] Discretized Integrated Gradients for Explaining Language Models.
>
> **Table R4: Results on R1-Distill-Qwen-1.5B using different interpolation steps $J$.**
> | Models                                    | Acc./Pass@1 | Length |
> |-------------------------------------------|-------------|--------|
> | **Greedy Decoding**                       |             |        |
> | R1-Distill-Qwen-1.5B                      | 36.7        | 20244  |
> | Full CoT SFT                              | 44.8        | 16520  |
> | Segment Selective SFT (IG J=50)           | 46.9        | 13506  |
> | Segment Selective SFT (IG J=20)           | 45.7        | 15288  |
> |                                           |             |        |
> | **Temperature Sampling**                  |             |        |
> | R1-Distill-Qwen-1.5B                      | 49.5        | 9791   |
> | Full CoT SFT                              | 50.9        | 10043  |
> | Segment Selective SFT (IG J=50)           | 51.7        | 9388   |
> | Segment Selective SFT (IG J=20)           | 51.3        | 9690   |

---

> ### Author Response · Authors · 2025-11-27
> **Response to Reviewer sHrL (2/2)**
>
> > Q3: Most of the experiment is conducted under Qwen-family framework, which leave the possibility that this method benefits from the model's characteristics cannot be completely ruled out. Experiments under different model architecture is suggested.
>
> Thank you for the suggestion. To verify that our method is not tied to Qwen architecture, we additionally train **LLaMA3.1-8B-Instruct** using the LIMO dataset. As shown in Table R3, our approach also yields clear performance gains in both accuracy and output efficiency.
>
> **Table R3: Comparison results on LLaMA3.1-8B-Instruct.**
> | Models                   | Acc./Pass@1 | Length |
> |--------------------------|-------------|--------|
> | **Greedy Decoding**      |             |        |
> | LLaMA3.1-8B-Instruct      | 24.0        | 9691   |
> | Full CoT SFT              | 30.4        | 17474  |
> | Segment Selective SFT     | 33.8        | 16005  |
> | **Temperature Sampling**  |             |        |
> | LLaMA3.1-8B-Instruct      | 23.5        | 4161   |
> | Full CoT SFT              | 33.0        | 15067  |
> | Segment Selective SFT     | 33.5        | 14669  |
>
> ---
>
> > Q4: Is there more application scenarios could be employed by those IG metrics.
>
> Yes. The IG-based metrics can be applied beyond our current setting. For example, they can be used to (1) identify key reasoning content for generating concise, reader-friendly explanations, and (2) guide reinforcement learning by emphasizing policy updates on the most influential reasoning segments. These extensions offer promising avenues for improving both reasoning quality and training efficacy.

---

### Official Review · Reviewer_KP2C · 2025-10-31

**Soundness:** 4
**Presentation:** 4
**Contribution:** 3
**Rating:** 6
**Confidence:** 4

**Summary:**

This paper introduces a segment-level attribution framework for selectively learning from long chain-of-thought (CoT) traces produced by large reasoning models. The approach leverages integrated gradients to assign an attribution “strength” and “direction consistency” to each CoT segment, identifying those segments with substantial and mixed influence on answer prediction. Selective supervised fine-tuning is then applied on only the important segments while masking loss for uninformative parts.

Experiments on multiple models and mathematics/reasoning datasets show improved accuracy and output efficiency versus standard full-CoT fine-tuning. The paper is supported by careful empirical and ablation studies and provides a detailed analysis of redundancy in CoTs.

**Strengths:**

1. Evaluates both pruning-based and selective learning variants, tests random and prior-importance alternatives, and shows that selective learning is preferable to simple pruning, with the IG-based method outperforming confidence, entropy, and perplexity-based segment selection (Table 3, Page 8).

2. The use of normalized strength and moderate consistency to select segments is justified theoretically and supported by empirical analysis (Section 3.1, Figures 2 and 3; the ablation study in Table 2, Page 8, directly supports the effectiveness of the joint criterion over alternatives).

3. Figures are used to substantiate both the intuition and the rigor (e.g., Figure 1 summarizes the distribution and accumulation of attribution across segments, while Figure 3 provides boxplots linking log perplexity, entropy, and BLEU similarity between important/unimportant segments).

4. Proposes segment-level attribution metrics based on integrated gradients—specifically, attribution strength and direction consistency—combined in a well-motivated and interpretable manner. The mathematical formulations (Equations on Pages 3–4) are clearly specified and grounded in attribution theory.

**Weaknesses:**

> Completeness of the chosen keywords for segmentation

Are the keywords used for segmentation comprehensive? How were the keywords selected? The generalizability of this paragraph segmentation method across different models (e.g., DeepSeek-R1 and Qwen2.5-7B) requires further discussion.

> Limited theoretical analysis of attribution methods

Although the attribution metrics are intuitive and based on empirical motivations, there is insufficient discussion of their theoretical limitations or the potential for attribution leakage. For instance, integrated gradients may produce artifacts for redundant input or model features, which could falsely elevate or lower paragraph scores. There is no thorough analysis or mitigation (e.g., using SmoothGrad) to reduce variance or noise in the attribution process.

> Threshold sensitivity and generalization

The critical thresholds for attribution strength and consistency (i.e., $\tau$ and $\beta$) are set via greedy search. However, the analysis of their stability or transferability to other datasets or inference types beyond mathematical QA is limited. Figure 5 (on page 15) addresses threshold effects, but only for one model/dataset, and does not provide a detailed analysis of generalization robustness.

> Insufficient evaluation metrics for diversity sampling.

In the results presented in Table 1, the pass@k value for temperature sampling is not provided, which is insufficient to fully capture the performance of the method across different benchmarks.

> Limited performance improvement

As shown in Table 1, for certain benchmarks, the accuracy after segment-selective SFT training is either lower or unchanged compared to the full CoT.

**Questions:**

1. Is the method for selecting keywords reasonable? How are the keywords chosen? Does this method apply to various models of different types?

2. Hyperparameter search in Sec. 3.2: The hyperparameter search method is conducted only for a single model and dataset. If the model or dataset is changed, is it necessary to re-search the hyperparameters? Given that the hyperparameter search requires first using the model to calculate the accuracy gain on the tasks, does this result in significant cost?

3. Applicability to other domains: Is this method still reasonable in other domains, such as code or general domains?

4. Failure case analysis: Have you reviewed any cases where your segmentation choices led to a performance drop in the model? Could such failures encourage further improvements to the masking strategy’s features?

---

> ### Author Response · Authors · 2025-11-27
> **Response to Reviewer KP2C (1/3)**
>
> Thank you for your detailed comments and constructive feedback! Below are our responses to each of them.
>
> > W1&Q1: Is the method for selecting keywords reasonable? How are the keywords chosen? Does this method apply to various models of different types?
> 1. **How we select the transition keywords.** Most of our transition keywords are directly adopted from Lu et al. (2025) [1], including“\n\nWait”, “\n\nAlternatively”, “\n\nHowever”, “\n\nNot sure”, “\n\nGoing back”, “\n\nBacktrack”, “\n\nTrace back”, and “\n\nAnother”.We prepend “\n\n” to each keyword because many LRMs naturally structure their reasoning output with paragraph breaks, making this prefix a reliable delimiter for identifying reasoning segments. Our keywords differ from [1] in two ways. First, instead of the broad keyword “But”, we use more specific variants including “\n\nBut wait”, “\n\nBut alternatively”, “\n\nBut just to”. This change is motivated by empirical observations that LRMs, especially smaller distilled models like R1-Distill-Qwen2.5-7B, tend to use “But” as conversational fillers rather than a meaningful reasoning transition. Second, we exclude fillers such as “Hmm” and “Hmmm”, which we also find to be frequently used as verbal habits instead of genuine segment boundaries. These design choices aim to balance generalizability across LRMs while ensuring that segmented reasoning units are meaningful rather than overly fragmented.
> 2. **Additional experiment with different keywords.** To directly address the reviewer’s concern, we also experiment with using the exact keyword list from [1]. We apply these keywords to segment the original LIMO CoTs and retrain R1-Distill-Qwen2.5-1.5B. This results in much finer-grained segmentation, increasing the average number of segments from 22.5 to 33.5 per reasoning trace. The selective SFT results (Table R1) show that this finer segmentation leads to higher accuracy and more concise outputs. We attribute this to that LIMO traces are generated by higher-quality models (e.g., DeepSeek-R1, QwQ-32B), whose reasoning contains fewer meaningless fillers. In such cases, finer segmentation yields more precise identification of important reasoning units.
> 3. **Keyword generalizability across models.** As implemented in the original paper, we apply our keyword set to segment both LIMO CoTs (mostly generated by DeepSeek-R1 and QwQ-32B) and self-generated CoTs (by smaller R1-Distill-Qwen2.5-7B) and achieve performance gain. In both cases, segment-based selective SFT consistently improves model performance. This demonstrates that our keyword selection is applicable across models.
>
> [1] Retro-Search: Exploring Untaken Paths for Deeper and Efficient Reasoning.
>
> **Table R1: Results on R1-Distill-Qwen-1.5B using new transition keywords.**
> | Models                                    | Acc./Pass@1 | Length |
> |-------------------------------------------|-------------|--------|
> | **Greedy Decoding**                       |             |        |
> | R1-Distill-Qwen-1.5B                      | 36.7        | 20244  |
> | Segment Selective SFT (Original Keywords) | 46.9        | 13506  |
> | Segment Selective SFT (New Keywords)      | 51.5        | 12646  |
> |                                           |             |        |
> | **Temperature Sampling**                  |             |        |
> | R1-Distill-Qwen-1.5B                      | 49.5        | 9791   |
> | Segment Selective SFT (Original Keywords) | 51.7        | 9388   |
> | Segment Selective SFT (New Keywords)      | 51.5        | 9167   |

---

> ### Author Response · Authors · 2025-11-27
> **Response to Reviewer KP2C (2/3)**
>
> > W2: Although the attribution metrics are intuitive and based on empirical motivations, there is insufficient discussion of their theoretical limitations or the potential for attribution leakage. For instance, integrated gradients may produce artifacts for redundant input or model features, which could falsely elevate or lower paragraph scores. There is no thorough analysis or mitigation (e.g., using SmoothGrad) to reduce variance or noise in the attribution process.
>
> - Thank you for raising this point. We agree that attribution methods including Integrated Gradients (IG) can suffer from noise and attribution leakage (i.e., gradient-based attributions spilling over from causally relevant tokens to irrelevant tokens due to attention diffusion or interpolation artifacts. However, our method aggregates IG values at the segment level instead of the token level, which naturally smooths token-level noise and reduces susceptibility to small attribution artifacts, making the segment importance ranking stable and robust in practice.
> - To further explore noise mitigation, we additionally apply SmoothGrad (3 samples, noise_std = 0.1, $J = 20$) and compare it with standard IG using $J = 20$ and $J = 50$. As shown in Table R5, SmoothGrad provides modest improvements over IG with the same number of interpolation steps, while IG with $J = 50$ remains the strongest overall. These results indicate that (1) attribution noise does not substantially affect our segment selection, and (2) our framework is compatible with noise-reduction techniques when desired.
>
> **Table R5: Results on R1-Distill-Qwen-1.5B using different IG calculations.**
> | Models                                    | Acc./Pass@1 | Length |
> |-------------------------------------------|-------------|--------|
> | **Greedy Decoding**                       |             |        |
> | R1-Distill-Qwen-1.5B                      | 36.7        | 20244  |
> | Segment Selective SFT (IG J=50)           | 46.9        | 13506  |
> | Segment Selective SFT (IG J=20)           | 45.7        | 15288  |
> | Segment Selective SFT (SmoothGrad J=20)   | 46.0        | 13947  |
> |                                           |             |        |
> | **Temperature Sampling**                  |             |        |
> | R1-Distill-Qwen-1.5B                      | 49.5        | 9791   |
> | Segment Selective SFT (IG J=50)           | 51.7        | 9388   |
> | Segment Selective SFT (IG J=20)           | 51.3        | 9690   |
> | Segment Selective SFT (SmoothGrad J=20)   | 51.3        | 9387   |
>
> ---
>
> > W3 & Q2: Hyperparameter search in Sec. 3.2: The hyperparameter search method is conducted only for a single model and dataset. If the model or dataset is changed, is it necessary to re-search the hyperparameters? Given that the hyperparameter search requires first using the model to calculate the accuracy gain on the tasks, does this result in significant cost?
> - Although our hyperparameter search is conducted only using R1-Distill-Qwen2.5-7B, the resulting configuration generalizes well across different models and different sources of CoT supervision within the LIMO dataset. Specifically, we use the same configuration ($\tau=0.7$ and $\beta=0.8$) to obtain important segments for training Qwen2.5-7B-Instruct, LLama3.1-8B-Instruct, R1-Distill-Qwen2.5-1.5B and R1-Distill-Qwen2.5-7B, and use both provided and self-generated CoT supervisions, all of which achieve consistent improvements.
> - For new datasets, we can efficiently identify appropriate thresholds by sampling a small subset of training examples and applying the same lightweight hyper-parameter search procedure (maximizing the confidence gap between important vs. unimportant segments). Specifically, generating 32 sampled answers after each segment for all LIMO data (817 instances) roughly cost 2 (GPU$\times$hours). This makes the framework practical even when adapting to different domains and tasks.

---

> ### Author Response · Authors · 2025-11-27
> **Response to Reviewer KP2C (3/3)**
>
> > W4: In the results presented in Table 1, the pass@k value for temperature sampling is not provided, which is insufficient to fully capture the performance of the method across different benchmarks.
>
> Thank you for the suggestion. Since we sample 6 responses per instance for MATH500, GPQA-Diamond, Minerva, and OlympiadBench, and 32 responses per instance for smaller benchmarks (AMC23 and AIME24), we compute pass@6 for all models across all benchmarks. The updated pass@6 results are presented in Table R6. We observe that Segment Selective SFT consistently improves over both the backbone model and full CoT SFT in term of pass@6.
>
> **Table R6: Pass@6 results on different backbone models.**
> | Models                | R1-Distill-Qwen-1.5B | R1-Distill-Qwen-7B | Qwen2.5-7B-Instruct |
> |-----------------------|------------------------|----------------------|----------------------|
> | Backbone Model     | 71.1                   | 79.2                 | 60.2                    |
> | Full CoT SFT      | 72.4                   | 79.2                 | 66.0                    |
> | Segment Selective SFT | **73.2**          | **80.0**             | **66.5**             |
>
> ---
>
> > W5: Limited performance improvement. As shown in Table 1, for certain benchmarks, the accuracy after segment-selective SFT training is either lower or unchanged compared to the full CoT.
> 1. Our method jointly improves both output efficiency and accuracy, which is difficult to achieve simultaneously in prior work. Under greedy decoding, segment-selective SFT delivers significant overall accuracy gains together with substantial token-length reduction. In contrast, improvements under temperature sampling are relatively smaller, as the randomness from multiple sampled trajectories tends to smooth out and partially offset the benefits introduced during training.
> 2. Small performance fluctuations across benchmarks are common and expected, as different datasets exhibit different characteristics. The few cases where segment-selective SFT underperforms full CoT typically fall into two categories: (1) benchmarks such as MATH500 and AMC23 where the baseline models already perform strongly and the generated sequences are not very long; and (2) out-of-domain benchmarks where full CoT training itself does not improve (and sometimes even reduces) performance compared to the baseline. In these scenarios, fully consistent improvements over full CoT are difficult to guarantee.
>
> ---
>
> > Q3: Applicability to other domains: Is this method still reasonable in other domains, such as code or general domains?
>
> Thank you for the question. While we did not include experiments on code or broader general-domain tasks due to space and resource constraints, we believe our method is conceptually applicable to these domains. The key assumption is not domain-specific that LRMs tend to generate redundant reasoning traces and different parts of a reasoning trace contribute unequally to final correctness. Segment-selective SFT operates by identifying and reweighting influential reasoning segments based on model behaviors rather than domain features, making it naturally domain-agnostic and compatible with tasks such as code generation or general reasoning, where long intermediate chains also occur. Extending segment-selective SFT to these domains is an important direction for future work, and we expect the approach to transfer well given its domain-independent design.
>
> ---
>
> > Q4: Failure case analysis: Have you reviewed any cases where your segmentation choices led to a performance drop in the model? Could such failures encourage further improvements to the masking strategy’s features?
>
> We would like to clarify that in our pipeline, segmentation serves as the first preprocessing step that divides long reasoning traces into manageable units. Then we need to obain IG-based segment importance to guide selective SFT across all training data. Therefore, we cannot directly attribute performance drops on evaluation data to segmentation errors in the training data, since segmentation alone does not influence the supervision signal. Additionally, as shown in our response to W1&Q1, our method generalizes well when using different keyword set for segmentation, suggesting robustness to reasonable segmentation variations. We therefore did not observe segmentation-specific failure cases that directly led to performance degradation.

---

### Official Review · Reviewer_2YhS · 2025-11-01

**Soundness:** 3
**Presentation:** 3
**Contribution:** 3
**Rating:** 6
**Confidence:** 5

**Summary:**

They introduces a method to improve learning from CoT reasoning sequences in LRMs. It identifies which segments of a reasoning trace contribute to the correct answer using segment-level attributions derived from Integrated Gradients. Segments are evaluated for attribution strength and direction consistency, and only important segments are used for supervised fine tuning.

The key contribution is demonstrating that models can learn more effectively by focusing on the reasoning steps that truly matter.

**Strengths:**

good analytical experiments
reliance on gradient based method rather LLM as judge
The use of Attribution Strength combined with Attribution Direction Consistency provides a mechanism to distinguish critical reasoning from superficial or shallow content
efficiency gains in terms of reasoning length (while having high/comparable accuracy)
Comprehensive ablation studies

**Weaknesses:**

- Focus on only one LLM (Qwen)
- High computational cost for importance calculation (J=50)
- Sensitivity to hyperparameter selection: the performance of the framework hinges on two crucial, empirically determined thresholds: τ (strength threshold) and β (consistency threshold). The hyperparameter search indicates that model performance is sensitive to these values, as choosing a higher τ introduced more false positives and negatively impacted training performance. The selection process (maximizing the difference in confidence gain between important/unimportant segments) is justified, but sensitivity remains a potential concern when moving to drastically different models or tasks.
- The segment partitioning relies on a pre-defined set of common transition keywords (e.g., “\n\nWait”, “\n\nAlternatively”)
- best performance is not indicated in the tables

**Questions:**

1. you mention that: "We estimate the correct answer confidence by multiple temperature-sampled generations". how do you calculate confidence?
2. Why did you choose to work in segment level? why not token level?
3. you mention that: "Extremely consistent attribution directions (all positive or all negative) may indicate shallow reasoning (e.g., dispensable explanation of final answers) while moderate consistency often reflect reflective and critical reasoning patterns where the model explores different possibilities, refines its understanding, and eliminates incorrect content, which are more valuable for learning effective reasoning strategies.". How do you claim this?
4. Could you please provide a quantification of the computational cost of the IG attribution step? Specifically, relative to the total time required for a standard Full CoT SFT epoch, how much overhead does the segment attribution calculation add?
5. The IG values were calculated using R1-Distill-Qwen2.5-7B. Did you explore how effective the segments identified by this 7B model are when fine-tuning much larger or smaller models, or models from entirely different families (i.e., treating the attribution model as an external tool)? This would help demonstrate the robustness of the identified importance definitions beyond a single model lineage.

---

> ### Author Response · Authors · 2025-11-27
> **Response to Reviewer 2YhS (1/3)**
>
> Thank you for your detailed comments and constructive feedback! Below are our responses to each of them.
>
> > W1: Focus on only one LLM (Qwen)
> - Although previous experiments in the paper are conducted on the Qwen family, they already cover both instruct model and distilled reasoning models across different parameter scales.
> - To further address the reviewer's concern, we additionally train **LLama3.1-8B-Instruct** using the LIMO dataset. As shown in Table R3, Segment Selective SFT still achieves performance improvement, demonstrating that our method generalizes well to a different model family.
>
> **Table R3: Comparison results on LLaMA3.1-8B-Instruct.**
> | Models                   | Acc./Pass@1 | Length |
> |--------------------------|-------------|--------|
> | **Greedy Decoding**      |             |        |
> | LLaMA3.1-8B-Instruct      | 24.0        | 9691   |
> | Full CoT SFT              | 30.4        | 17474  |
> | Segment Selective SFT     | 33.8        | 16005  |
> | **Temperature Sampling**  |             |        |
> | LLaMA3.1-8B-Instruct      | 23.5        | 4161   |
> | Full CoT SFT              | 33.0        | 15067  |
> | Segment Selective SFT     | 33.5        | 14669  |
>
> ---
>
> > W2&Q4: High computational cost for importance calculation (J=50). Could you please provide a quantification of the computational cost of the IG attribution step? Specifically, relative to the total time required for a standard Full CoT SFT epoch, how much overhead does the segment attribution calculation add?
> - To ensure high-fidelity IG calculation, we use R1-Distill-Qwen2.5-7B with $J=50$ interpolation steps. This costs roughly 7 (GPU$\times$hours) compared to the 8 (GPU$\times$hours) needed for a full SFT run of the same 7B model.
> - The cost can be further reduced by decreasing the interpolation step $J$. Our additional experiments (in Table R4) show that using a smaller $J$ ($J=20$) still yields strong attribution quality with a minimal performance drop while still outperforming full CoT SFT. In this setting, IG computation becomes much more efficient, requiring only about 3 GPU-hours.
>
> **Table R4: Results on R1-Distill-Qwen-1.5B using different interpolation steps $J$.**
> | Models                                    | Acc./Pass@1 | Length |
> |-------------------------------------------|-------------|--------|
> | **Greedy Decoding**                       |             |        |
> | R1-Distill-Qwen-1.5B                      | 36.7        | 20244  |
> | Full CoT SFT                              | 44.8        | 16520  |
> | Segment Selective SFT (IG J=50)           | 46.9        | 13506  |
> | Segment Selective SFT (IG J=20)           | 45.7        | 15288  |
> |                                           |             |        |
> | **Temperature Sampling**                  |             |        |
> | R1-Distill-Qwen-1.5B                      | 49.5        | 9791   |
> | Full CoT SFT                              | 50.9        | 10043  |
> | Segment Selective SFT (IG J=50)           | 51.7        | 9388   |
> | Segment Selective SFT (IG J=20)           | 51.3        | 9690   |
>
> ---
>
> > W3: Sensitivity to hyperparameter selection: the performance of the framework hinges on two crucial, empirically determined thresholds: τ (strength threshold) and β (consistency threshold). The hyperparameter search indicates that model performance is sensitive to these values, as choosing a higher τ introduced more false positives and negatively impacted training performance. The selection process (maximizing the difference in confidence gain between important/unimportant segments) is justified, but sensitivity remains a potential concern when moving to drastically different models or tasks.
>
> - Although different threshold configurations can influence training performance, our hyperparameter search process reliably identifies suitable values for training. More importantly, we find that the selected thresholds generalize well across different models and different sources of CoT supervision on the LIMO dataset. Specifically, we use the same configuration ($\tau=0.7$ and $\beta=0.8$) to obtain important segments for training Qwen2.5-7B-Instruct, LLama3.1-8B-Instruct, R1-Distill-Qwen2.5-1.5B and R1-Distill-Qwen2.5-7B, and use both provided and self-generated CoT supervisions, all of which achieve consistent improvements.
> - For new datasets, we can efficiently identify appropriate thresholds by sampling a small subset of training data and applying the same lightweight hyper-parameter search procedure (maximizing the confidence gap between important vs. unimportant segments). This makes the framework practical and robust even when adapting to different domains and tasks.

---

> ### Author Response · Authors · 2025-11-27
> **Response to Reviewer 2YhS (2/3)**
>
> > W4: The segment partitioning relies on a pre-defined set of common transition keywords (e.g., “\n\nWait”, “\n\nAlternatively”).
>
> Most of our transition keywords are directly adopted from [1], with some modifications and filters applied to improve generalizability across different LLMs. To investigate our method's  effectiveness regarding different keywords, we additionally adopt the exact keyword set  from [1]. We re-segment the original LIMO CoTs using this keyword list and retrain R1-Distill-Qwen2.5-1.5B. This results in much finer-grained segmentation, increasing the average number of segments from 22.5 to 33.5 per reasoning trace. The results in Table R1 illustrate that our method remains robust under this alternative keyword set, and more fine-grained selective SFT can yield better performance and more efficient output.
>
> [1] Retro-Search: Exploring Untaken Paths for Deeper and Efficient Reasoning.
>
> **Table R1: Results on R1-Distill-Qwen-1.5B using new transition keywords.**
> | Models                                    | Acc./Pass@1 | Length |
> |-------------------------------------------|-------------|--------|
> | **Greedy Decoding**                       |             |        |
> | R1-Distill-Qwen-1.5B                      | 36.7        | 20244  |
> | Segment Selective SFT (Original Keywords) | 46.9        | 13506  |
> | Segment Selective SFT (New Keywords)      | 51.5        | 12646  |
> |                                           |             |        |
> | **Temperature Sampling**                  |             |        |
> | R1-Distill-Qwen-1.5B                      | 49.5        | 9791   |
> | Segment Selective SFT (Original Keywords) | 51.7        | 9388   |
> | Segment Selective SFT (New Keywords)      | 51.5        | 9167   |
>
> ---
>
> > W5: best performance is not indicated in the tables
>
> Thank you for the suggestion. We have updated all relevant tables and highlight the best performance in bold for easier comparison.
>
> ---
>
> > Q1: you mention that: "We estimate the correct answer confidence by multiple temperature-sampled generations". How do you calculate confidence?
>
> We apologize for the confusion. As described in lines 240–243, "We estimate the correct answer confidence by multiple temperature-sampled generations (i.e., 32 samples) to calculate the frequency of correct answers." In other words, the model’s confidence on the correct answer is defined as **the probability that it generates the correct answer**. However, directly computing this probability from the model’s token-level output distribution after each segment can be unreliable because (1) the same answer may have multiple valid surface forms (especially numeric answers), and (2) formatting tokens may affect the computed probability. To avoid these issues, we estimate confidence empirically as **the fraction of sampled generations that produce the correct answer**.
>
> ---
>
> > Q2: Why did you choose to work in segment level? why not token level?
>
> - As stated in lines 49–50, token-level analysis neglects **semantic integrity and fails to interpret redundancy in terms of meaningful reasoning units**. Selective training on incoherent token fragments would impact the fluency and coherence of the model’s generated reasoning.
> - In addition, IG attribution inevitably contains noise or attribution leakage, i.e., gradient-based attributions spilling over from causally relevant tokens to irrelevant tokens due to attention diffusion or interpolation artifacts. By aggregating IG scores at the segment level rather than the token level, our method naturally **smooths out token-level noise and becomes less sensitive to small attribution artifacts**, leading to more stable and reliable importance estimates.
> - Finally, the ablation results in Table 2 confirm that selective learning on High-Abs-IG Tokens or High-Orig-IG Tokens performs worse than our segment-level approach, and even does not outperform full CoT supervision. These findings demonstrate that token-level selection is insufficient, and motivate our choice to operate at the segment level.

---

> ### Author Response · Authors · 2025-11-27
> **Response to Reviewer 2YhS (3/3)**
>
> > Q3: you mention that: "Extremely consistent attribution directions (all positive or all negative) may indicate shallow reasoning (e.g., dispensable explanation of final answers) while moderate consistency often reflect reflective and critical reasoning patterns where the model explores different possibilities, refines its understanding, and eliminates incorrect content, which are more valuable for learning effective reasoning strategies.". How do you claim this?
> 1. **Evidence from positional distribution and confidence analysis.** As shown in the positional distribution analysis in Appendix A, *72% of high-consistency unimportant segments appear after the decision segment* (i.e., the first segment where the model derives the correct answer). Combined with Figure 2 that shows that *high-consistency segments typically induce minimal improvement in correct-answer confidence*, this suggests that such segments mostly perform formalized verification and reinforcement of an already-found answer rather than introducing deeper critical examination.
> 2. Besides, extremely consistent attribution directions specifically include two patterns: (1) Most tokens are positive, and (2) Most tokens are negative.
>     - For case (1), we evaluate **how often the correct answer appears within such segments**. Positive segments with consistency > 0.8 mention the correct answer 1.50 times on average, whereas those with consistency < 0.5 mention it only 0.84 times. This supports the view that segments with most positive tokens tend to be shallow reasoning (e.g., dispensable explanation of final answers).
>     - For case (2), we measure **how many new unique numbers each segment introduces as a proxy for whether the model is exploring new possibilities**. Negative segments with consistency > 0.8 introduce only 2.0 new unique numbers on average, while those with consistency < 0.5 introduce 3.7. This indicates that segments with most negative tokens convey less exploratory information, whereas moderately consistent negative segments provide richer elimination-based reasoning.
>
> ---
>
> > Q5: The IG values were calculated using R1-Distill-Qwen2.5-7B. Did you explore how effective the segments identified by this 7B model are when fine-tuning much larger or smaller models, or models from entirely different families (i.e., treating the attribution model as an external tool)? This would help demonstrate the robustness of the identified importance definitions beyond a single model lineage.
>
> Yes. In our current pipeline, we use R1-Distill-Qwen2.5-7B as the attribution model to compute IG values and identify important segments from both LIMO-provided CoTs and self-generated CoTs by R1-Distill-Qwen2.5-7B. We then conduct selective SFT on these identified segments for models of different sizes and families, including R1-Distill-Qwen2.5-1.5B, R1-Distill-Qwen2.5-7B, Qwen2.5-7B-Instruct and LLama3.1-8B-Instruct (The first three are shown in Table 1 in the original submission and the last one is added in Table R3). These results (Table 1 in the main paper and Table R3 in the rebuttal) consistently show performance improvements, demonstrating the robustness of our identified important segments.

---

### Official Review · Reviewer_Jvus · 2025-11-01

**Soundness:** 3
**Presentation:** 3
**Contribution:** 4
**Rating:** 8
**Confidence:** 4

**Summary:**

The paper proposes a segment-level selection method for CoT reasoning that identifies and trains only those segments that meaningfully contribute to correct answer generation (for SFT). Using Integrated Gradients (IG), the authors measure each segment’s attribution to the final answer and summarise this using two metrics: Attribution Strength and Attribution Direction Consistency. An interesting design choice is that Attribution Strength is defined using the absolute IG values (rather than the signed values), and that the method selects segments with moderate, rather than high, Direction Consistency. The stated motivation is that (i) even reasoning that initially explores incorrect directions may still be useful later, and therefore should not be discarded purely because it has negative IG, and (ii) segments with very high consistency may reflect shallow or overly biased reasoning rather than careful reflection. Across multiple models (R1-Distill-Qwen2.5-1.5B/7B, Qwen2.5-7B-Instruct) and datasets (LIMO, MATH500, AIME, etc.), the proposed approach improves accuracy while reducing output length.

**Strengths:**

- It is interesting to see the use of IG at the segment level to score the importance of different parts of the CoT. The definitions of Attribution Strength and Attribution Direction Consistency are thoughtful in that they are not purely driven by raw gradient magnitude, but instead attempt to capture properties specific to reasoning structure.

- Through extensive experiments, the authors present hyperparameter search, ablations, and comparisons with alternative importance measures.

- The writing is clear and the figures effectively illustrate the ideas and motivation.

**Weaknesses:**

- The argument for using attribution strength based on the absolute value of IG may need to be strengthened. The paper motivates this choice by noting that exploratory reasoning can sometimes have negative IG values, and that such reasoning should not be thrown away. However, consider a segment that overall shows strongly negative IG values and only moderate consistency. Is it necessarily the case that such a segment corresponds to “necessary exploratory reasoning” (lines 155–156)? While useful exploratory reasoning may indeed yield negative IG values, it is less clear that the reverse implication always holds. In other words, segments with strong negative attribution are not guaranteed to be beneficial; they may simply reflect confident but unhelpful reasoning.

**Questions:**

- How were the segmentation keywords in Appendix C.2 selected? The transition keywords used here appear to differ from the list in Lu et al. (2025). Have the authors evaluated the effect of varying this keyword list on the final results?

- In Table 2, could the authors provide an additional comparison for the “High Strength Segments” condition where segments with very large negative IG attribution are excluded? This would help clarify whether strongly negative-attribution segments are genuinely useful or whether they can be safely filtered out.

---

> ### Author Response · Authors · 2025-11-27
> **Response to Reviewer Jvus (1/2)**
>
> Thank you for your detailed comments and constructive feedback! Below are our responses to each of them.
>
> > Q1: How were the segmentation keywords in Appendix C.2 selected? The transition keywords used here appear to differ from the list in Lu et al. (2025). Have the authors evaluated the effect of varying this keyword list on the final results?
>
> 1. **How we select the transition keywords.** Most of our transition keywords are directly adopted from Lu et al. (2025) [1], including `“\n\nWait”, “\n\nAlternatively”, “\n\nHowever”, “\n\nNot sure”, “\n\nGoing back”, “\n\nBacktrack”, “\n\nTrace back”, and “\n\nAnother”`. We prepend “\n\n” to each keyword because many LRMs naturally structure their reasoning output with paragraph breaks, making this prefix a reliable delimiter for identifying reasoning segments. Our keywords differ from [1] in two ways. First, instead of the broad keyword “But”, we use more specific variants including `“\n\nBut wait”, “\n\nBut alternatively”, “\n\nBut just to”`. This change is motivated by empirical observations that LRMs, especially smaller distilled models like R1-Distill-Qwen2.5-7B, tend to use “But” as conversational fillers rather than a meaningful reasoning transition. Second, we exclude fillers such as “Hmm” and “Hmmm”, which we also find to be frequently used as verbal habits instead of genuine segment boundaries. These design choices aim to balance generalizability across LRMs while ensuring that segmented reasoning units are meaningful rather than overly fragmented.
> 2. **Additional experiment with different keywords.** To directly address the reviewer’s concern, we also experiment with using the exact keyword list from [1]. We apply these keywords to segment the original LIMO CoTs and retrain R1-Distill-Qwen2.5-1.5B. This results in much finer-grained segmentation, increasing the average number of segments from 22.5 to 33.5 per reasoning trace. The selective SFT results (Table R1) show that this finer segmentation leads to higher accuracy and more concise outputs. We attribute this to that LIMO traces are generated by higher-quality models (e.g., DeepSeek-R1, QwQ-32B), whose reasoning contains fewer meaningless fillers. In such cases, finer segmentation yields more precise identification of important reasoning units.
>
> We will incorporate all implementation details and the new experimental results into the revised manuscript.
>
> [1] Retro-Search: Exploring Untaken Paths for Deeper and Efficient Reasoning.
>
> **Table R1: Results on R1-Distill-Qwen-1.5B using new transition keywords.**
> | Models                                    | Acc./Pass@1 | Length |
> |-------------------------------------------|-------------|--------|
> | **Greedy Decoding**                       |             |        |
> | R1-Distill-Qwen-1.5B                      | 36.7        | 20244  |
> | Segment Selective SFT (Original Keywords) | 46.9        | 13506  |
> | Segment Selective SFT (New Keywords)      | 51.5        | 12646  |
> |                                           |             |        |
> | **Temperature Sampling**                  |             |        |
> | R1-Distill-Qwen-1.5B                      | 49.5        | 9791   |
> | Segment Selective SFT (Original Keywords) | 51.7        | 9388   |
> | Segment Selective SFT (New Keywords)      | 51.5        | 9167   |

---

> ### Author Response · Authors · 2025-11-27
> **Response to Reviewer Jvus (2/2)**
>
> > W1&Q2: The argument for using attribution strength based on the absolute value of IG may need to be strengthened. The paper motivates this choice by noting that exploratory reasoning can sometimes have negative IG values, and that such reasoning should not be thrown away. However, consider a segment that overall shows strongly negative IG values and only moderate consistency. Is it necessarily the case that such a segment corresponds to “necessary exploratory reasoning” (lines 155–156)? While useful exploratory reasoning may indeed yield negative IG values, it is less clear that the reverse implication always holds. In other words, segments with strong negative attribution are not guaranteed to be beneficial; they may simply reflect confident but unhelpful reasoning. In Table 2, could the authors provide an additional comparison for the “High Strength Segments” condition where segments with very large negative IG attribution are excluded? This would help clarify whether strongly negative-attribution segments are genuinely useful or whether they can be safely filtered out.
> 1. **Why we use absolute IG values.** Our goal is to measure attribution at the segment level. If we use raw IG (with signs), positive and negative token attributions within the same segment may cancel out, obscuring the magnitude of that segment’s true contribution. We also cannot simply discard negative tokens and aggregate only positive tokens, because many negative tokens reflect useful elimination-based exploration. Table 2 already shows that absolute token-level IG values provide a stronger signal than raw IG. Furthermore, 67% of the first reasoning segments has an overall negative raw IG (with a mean value of –0.0095, close to the 20th percentile of all negative segment IGs), yet these initial segments are unquestionably essential for problem framing. This motivates using the absolute IG values as a more faithful segment-level importance measure.
> 2. **Whether strongly negative segments are useful.** Your question about whether the reverse implication holds (i.e., whether a segment with strong negative IG is always useful) is insightful. The answer is no. Strongly negative segments fall into two categories. (1) Strongly negative + high consistency: segments with strongly negative IG values and also high consistency (most tokens are negative) are typically unhelpful and are already removed by our consistency-based filter. (2) Strongly negative + moderate consistency: segments with strongly negative overall IG but only moderate consistency is necessary and usually correspond to reflective/elimination reasoning. For example, many first segments have strongly negative IGs (their mean value near the 20th percentile of all negative IGs) but moderate consistency (≈0.41).
> 3. **Additional experiment: filtering strongly negative segments.** To directly address your question, we conduct an additional experiment. From the original “High-Strength Segments,”we remove segments whose raw IG values fall below a strong-negative threshold. We set this threshold at the 20th percentile of all negative IG values (≈−0.0097). As shown in Table R2, performance drops further after removing strongly negative segments due to incurred false negatives. These results reinforce that strongly negative IGs at the segment level does not necessarily imply irrelevance, and directly removing such segments can harm reasoning fidelity.
>
> **Table R2: High-Strength Segments results without strongly negative segments.**
> | Models                                      | Acc./Pass@1 | Length |
> |---------------------------------------------|-------------|--------|
> | **Greedy Decoding**                         |             |        |
> | R1-Distill-Qwen-1.5B                        | 36.7        | 20244  |
> | Segment Selective SFT                       | 46.9        | 13506  |
> | High Strength Segments                      | 46.9        | 14715  |
> | High Strength Segments w/o Strong Negatives | 46.4        | 14020  |
> | **Temperature Sampling**                    |             |        |
> | R1-Distill-Qwen-1.5B                        | 49.5        | 9791   |
> | Segment Selective SFT                       | 51.7        | 9388   |
> | High Strength Segments                      | 51.4        | 9466   |
> | High Strength Segments w/o Strong Negatives | 51.1        | 9553   |

---

### Author Response · Authors · 2025-12-04
**Summary of Rebuttal**

Dear Area Chair,

We sincerely appreciate the time and effort you and the reviewers have dedicated to the review process. We are writing to provide a brief summary of the rebuttal to assist you in your final evaluation.

First, we are grateful for the reviewers' recognition of our work's strengths, including the **well-motivated metrics and effective method** (`Jvus`, `2YhS`, `KP2C`,`sHrL`), **thorough analysis** (`2YhS`, `KP2C`),  **comprehensive experiments** (`Jvus`, `2YhS`, `KP2C`), and **good presentation** (`Jvus`, `2YhS`, `KP2C`,`sHrL`). These strengths are also reflected in the consistent and positive scores across specific criteria, including **Soundnes(3,3,4,3)**, **Presentation (3,3,4,3)**, and **Contribution (4,3,3,3)**.

During the rebuttal period, we have supplemented additional experiments and analyses to address reviewers' concerns, including:
- Explanation on keyword selection and experiments showing keyword generalizability. (`Jvus`, `2YhS`,`KP2C`)
- Experimental results on a different modal family (LLama). (`2YhS`, `sHrL`)
- Computational costs and performance of integrated gradient attribution under varying settings. (`2YhS`, `KP2C`, `sHrL`)
- Investigation into the impact of strongly negative segments. (`Jvus`)
- Clarification on the definition of confidence calculation. (`2YhS`,`sHrL`)
- Discussion on the cost and sensitivity of hyperparameter search. (`2YhS`,`KP2C`)
- Added pass@k results. (`KP2C`)
- Additional empirical evidence of shallow reasoning, explanation on why segment-level. (`2YhS`)

We hope this summary assists in your assessment. Thank you again for your dedication to the thorough review process.


Sincerely,

The Authors

---

### Meta-Review · Area_Chair_tdVa · 2026-01-06

**Summary:**

This paper proposes a segment-level selective learning framework for Chain-of-Thought (CoT) reasoning. The core idea is to identify and train only on reasoning segments that meaningfully contribute to the correct answer, thereby filtering out redundancy and shallow heuristics. The authors utilize Integrated Gradients (IG) to compute two novel metrics: "Attribution Strength" and "Direction Consistency." The method selects segments with high strength and moderate consistency (posited to represent reflective reasoning). Experiments primarily using the Qwen family (and Llama in the rebuttal) demonstrate that this approach improves accuracy on mathematical benchmarks while significantly reducing output length compared to full CoT fine-tuning.

**Reviewer Concerns:**

Addressed Concerns:
- Model Generalization: Reviewers 2YhS and sHrL expressed concern that the experiments were limited to the Qwen family. The authors addressed this effectively by conducting additional experiments on LLaMA-3.1-8B-Instruct, demonstrating consistent performance gains and proving the method is not architecture-specific.
- Keyword/Segmentation Sensitivity: Reviewers Jvus, 2YhS, and KP2C questioned the robustness of the specific keywords used for segmentation. The authors performed an ablation using an alternative keyword set from prior literature (Lu et al., 2025). The results showed the method remained robust (and even improved with finer granularity), alleviating concerns about heuristic brittleness.
- Computational Cost of IG: Reviewers 2YhS and sHrL raised concerns regarding the overhead of calculating Integrated Gradients (J=50). The authors provided a cost analysis and showed that reducing interpolation steps to J=20 significantly reduces computational time (to approx. 3 GPU-hours) with negligible performance loss.
- Attribution Metrics & Noise: Reviewer Jvus questioned the use of absolute IG values, and KP2C asked about attribution noise. The authors provided empirical evidence that removing "strongly negative" segments harms performance (justifying their absolute value approach) and compared their method against SmoothGrad, showing their approach is robust to noise.
- Evaluation Metrics: Reviewer KP2C requested Pass@k metrics for temperature sampling, which the authors provided in the rebuttal, showing consistent improvements.

Outstanding Concerns:
- Hyperparameter Sensitivity: Reviewers 2YhS and KP2C noted that the method relies on empirically determined thresholds ($\tau$ and $\beta$). While the authors argue these transfer well across the tested models, the requirement for a search process (maximizing confidence gaps) to tune these values for potentially new domains remains a slight friction point compared to parameter-free methods.
- Computational Overhead vs. SFT: While the authors reduced the cost, there is still a pre-processing overhead involving gradient computations compared to standard SFT.

**Reviewer Scores:**

- Reviewer Jvus (Score: 8): Likely to remain a 8. Very positive regarding the motivation and the thoroughness of the analysis. Satisfied with the justification for using absolute IG values.
- Reviewer 2YhS (Score: 6): Likely to remain a 6. Positive about the efficiency gains and method. Remained marginally above threshold, likely due to the inherent complexity of the pipeline compared to standard training, though they acknowledged the rebuttal addressed the specific questions.
- Reviewer KP2C (Score: 6): Likely to remain a 6. Positive on soundness and presentation. The rebuttal addressed their main concerns regarding keywords and pass@k metrics.
- Reviewer sHrL (Score: 6): Likely to remain a 6. Positive on the presentation and the definitions of the metrics. Satisfied with the clarification on confidence estimation and the addition of Llama experiments.

---

### Decision · Program_Chairs · 2026-01-26

Accept (Poster)